Climatic variability in Princess Elizabeth Land (East Antarctica) over the last 350
years
Alexey A. Ekaykin[1,2], Diana O. Vladimirova[1,2*], Vladimir Ya. Lipenkov[1] and Valérie Masson-
Delmotte[3]
1 – Climate and Environmental Research Laboratory, Arctic and Antarctic Research Institute, St.
Petersburg, Russia.
2 – Institute of Earth Sciences, Saint Petersburg State University, St. Petersburg, Russia.
3 - Laboratoire des Sciences du Climat et de l'Environnement - IPSL, UMR 8212, CEA-CNRS-
UVSQ-Université Paris Saclay, Gif sur Yvette, France.
* now at Center for Ice and Climate, Niels Bohr Institute, University of Copenhagen, Juliane
Maries Vej 30, 2100 Copenhagen Ø, Denmark.
*Correspondence to:* Alexey A. Ekaykin (ekaykin@aari.ru)
**Abstract**
We use isotopic composition (δD) data from 6 sites in Princess Elisabeth Land (PEL) in order to
reconstruct the air temperature variability in this sector of East Antarctica for the last 350 years.
First, we use the present-day instrumental mean annual surface air temperature data to
demonstrate that the studied region (between Russian research stations Progress, Vostok and
Mirny) is characterized by uniform temperature variability. We thus construct the stacked record
of the temperature anomaly for the whole sector for the period 1958-2015. A comparison of this
series with the Southern Hemisphere climatic indices shows that the short-term inter-annual
temperature variability is primarily governed by Antarctic Oscillation (AAO) and Interdecadal
Pacific Oscillation (IPO) modes of atmospheric variability. However, the low-frequency
temperature variability (with period > 27 years) is mainly related to the anomalies of Indian
Ocean Dipole (IOD) mode. Then we construct the stacked record of δD for the PEL for the
period 1654-2009 from individual normalized and filtered isotopic records obtained at 6 different
sites ('PEL2016' stacked record). We use a linear regression of this record and the stacked PEL
temperature record (with an apparent slope of 9±5.4 ‰ °C$^{-1}$) to convert 'PEL2016' into
temperature scale. Analysis of 'PEL2016' shows a 1±0.6 °C warming in this region over the last

three centuries, with a particularly cold period from mid-18[th] to mid-19[th] century. A peak of cooling occurred in the 1840s - a feature previously observed in other Antarctic records. We reveal that 'PEL2016' correlates with a low-frequency component of IOD. We suggest that the IOD mode influences the Antarctic climate by modulating the activity of cyclones that bring heat and moisture to Antarctica. We also compare 'PEL2016' with other Antarctic stacked isotopic records. This work is a contribution to PAGES and IPICS Antarctica 2k projects.

## 1 Introduction

While understanding the behavior of Antarctic climate system is crucial in context of the present-day global environmental changes, key gaps arise from limited observations. Prior to the International Geophysical Year epoch (1955-1957) the primary source of the climatic data are ice core records. Deep ice cores have provided a wealth of climatic and environmental information covering glacial-interglacial variations of the past 800,000 years (EPICA, 2004). However, the spatio-temporal characteristics of Antarctic climate variability of the most recent centuries remains poorly known or understood (Jones et al., in press;PAGES_2k_network, 2013).

The network of ice core records spanning the last centuries is distributed highly unevenly. A quite extensive coverage of some regions of Antarctica, such as West Antarctica (Kaspari et al., 2004) or Dronning Maud Land (Altnau et al., 2015;Oerter et al., 2000) contrasts with other regions that still remain poorly studied. As a result, attempts to reconstruct the climatic variability of the whole Antarctic continent (Jones et al., in press;PAGES_2k_network, 2013;Schneider et al., 2006;Frezzotti et al., 2013) are limited by the lack of available data.

In our previous work we summarized available isotopic data for the vicinity of Vostok Station in order to construct a robust stack climatic record over the past 350 years (Ekaykin et al., 2014). Here we present a new stacked climate record for Princess Elisabeth Land (PEL), the territory located between the Russian stations of Progress, Vostok and Mirny, East Antarctica. This record  is based on water stable isotope data from 6 sites, and spans the last 350 years (Fig. 1). We note a not perfect correlation between the stacked isotopic record and regional surface air temperature variations, underlying the fact that the isotopic content of precipitation is not simply a proxy of temperature, but rather a parameter that covary with the local climate in a manner similar to temperature (Steig et al., 2013).

We also highlight significant relationships between regional climate and large-scale modes of
variability of the Southern Hemisphere.
Section 2 describes our data and methods, and Section 3 is focused on these results and their
discussion, before a conclusion in Section 4.

**2 Methods**
2.1 Ice core data
In this study we use data from 6 individual records obtained in Princess Elisabeth Land (Figure
1, Table 1).
"105 km" (67.433 ºS and 93.383 ºE, time interval 1757-1987) is a 727-m ice core drilled in 1988
by specialists of St. Petersburg Mining Institute, about 105 km inland from Mirny station. The
isotopic content was measured late in the 1980s at Laboratoire des Sciences du Climat et de
l'Environnement (LSCE) with resolution of 1 m. In 2013, the upper 109 m of the core were re-
measured at Climate and Environmental Research Laboratory (CERL), with a depth resolution of
5 cm. This core is the only one where accumulation rate allows the annual layers to be preserved
in the snow thickness, so the core was dated by layer counting. The initial dating was then
adjusted using the reference horizon of the 1816 Tambora volcanic eruption, identified from
Electrical Conductivity Measurements (ECM) (Vladimirova and Ekaykin, 2014). As a result, a
record of annual accumulation rate is available.
"400 km" (69.95 ºS and 95.617 ºE, 1254-1987) refers to an ice core drilled in 1988 at the 400[th]
km from Mirny station, down to a 150 m depth. Isotopic measurements were performed at LSCE
on 1 m samples. The core was dated according to the simple Nye depth-age model, taking into
account the average accumulation rate at the drilling site (Lipenkov et al., 1998) and the density
profile of the core. The uncertainty of the dating, estimated with the Nye model, mainly comes
from the error of the accumulation rate estimate and is evaluated as about 10 %. As a result, no
record of annual accumulation rate is available.
"VRS 2013" (78.467 ºS and 106.84 ºE, 1654-2010) is a stack of 15 individual isotopic records
from snow pits and shallow cores recovered in the vicinity of Vostok Station (Ekaykin et al.,
2014). The data on temporal variability of snow accumulation rate is also available for this site.
"NVFL-1" (77.11 ºS and 95.072 ºE, 1711-1944) is a 18.3-m firn core drilled from the bottom of
a 2.5-m snow pit in 2008 close to the Dome B. The chronology was established using the firn
density data and the 1816 Tambora volcano ECM peak as a reference horizon.
"NVFL-3" (76.405 ºS and 102.167 ºE, 1978-2009) is a 3.1-m snow pit dug in 2010 in the
northern part of subglacial Lake Vostok. It is dated  based on snow stratigraphy and
identification of 1993 Pinatubo volcano peak in $SO_4^{2-}$ vertical profile. Chemical measurements
were performed at Limnological Institute of Russian Academy of Sciences, Irkutsk, Russia.
"PV-10" (72.805 ºS and 79.934 ºE, 1976-2009) is a 7.55-m firn core drilled in 2010 about 400
km inland from Progress Station. It was dated using firn density data and taking into account the
ECM peak associated with the 1993 deposition from the Pinatubo eruption.
We estimated the dating uncertainty by comparing age calculated using only firn density data
and average snow accumulation rate for a given site with age of the reference age markers and
came to a conclusion that the age errors do not exceed 10 %. For the reference years (1816 and
1993, where we have absolute dating), the error tends to zero. The largest error is expected for
the "400 km" series, where we do not have a reference age markers. However, if we use the
prominent 1840 cold event (see Section 3.3), observed in all records, as such a marker, then we
may estimate a relative dating error for this series as < 6%.
We also use the accumulation data from the site "200 km" (Fig. 1), spanning the period 1640-
1987, as published in (Ekaykin et al., 2000). The accumulation values from sites "150 km" and
"400 km" were corrected both for layer thinning with depth and for the advection of ice from
upstream of the glacier to account for the spatial gradient of the snow accumulation rate.

2.2 Stacked records
Fig. 2 displays the individual δD time-series from all 6 sites. Differences between mean values
reflect well-known differences in isotopic distillation along a gradient of inland elevation
(e.g.(Masson-Delmotte et al., 2008)). In order to investigate temporal variations only, we
calculated normalized values for each series using interval 1757-1944 as a reference period. As
for the short series (NVFL-3 and PV-10), they were normalized over 1978-2009 period, and then
the mean and variance of the normalized values were adjusted to those of the long series for the
corresponding period of time, in order to avoid an overestimated contribution of the short records
in the stacked series.
We then applied a rectangular-shaped low-pass filter to cut off the variability with periodicities
shorter than 27 years (i.e., frequencies > 0.037). (All spectral analyses and filtering were
performed with the use of Analyseries software (Paillard et al., 1996)). This is motivated by the
fact that one single record in inland Antarctica cannot provide  reliable climatic information on a
short-term time scale, due to a very low signal-to-nose ratio (Ekaykin et al., 2014) and non-
temperature effects on isotopes in precipitation including post-depositional alterations.
Moreover, the latter study also highlighted multi-decadal climatic variability in this sector of
central Antarctica, with a period of 30-50 years.
The normalized and filtered time series are displayed in Figure 3. Despite some common
features, this comparison shows significant discrepancies between individual records. One
reason for such mismatches may lie in age scale uncertainties. However, this hypothesis is ruled
out by the comparison of individual series around 1816 and 1993 (dates of firn layers containing
Tambora and Pinatubo volcanic eruption debris, denoted by vertical dashed lines in Figure 3),
when the relative dating error tends to zero: observed discrepancies do not solely arise from
chronological uncertainties alone. Alternatively, this mismatch may arise from a significant level
of noise even in the filtered series, and other than the local temperature-related controls of the
isotopic composition of precipitation.
In order to isolate the climatic signal from the noise, we constructed a stacked climatic record for
the PEL region, hereafter named PEL2016 (grey line in Figure 3). For a given year, the value of
this record consists of the average of the values of individual records available for this year.

2.3 Instrumental temperature data
A number of research stations have been established in the PEL area, as indicated in Fig. 1.
Unfortunately, most of them have very short (if any) meteorological records. Relatively long
records are available only for 5 stations: Australian station Davis (1957-1964 and 1969-2015),
Chinese station Zhong Shan (1989-2015), Russian stations Progress (1989, 1991 and 2003-
2015), Mirny (1956-2015) and Vostok (1958-2015 with gaps in 1962, 1994, 1996 and 2003).
The monthly data were downloaded from https://legacy.bas.ac.uk/met/READER/ (Turner et al., 2004)
and then the annual means were calculated.
The correlation between Progress, Zhong Shan and Davis annual mean temperature datasets,
located very close to each other, is 0.96-0.98 (note that only statistically significant correlation
coefficients with a confidence level > 95 % are reported in the paper, unless otherwise
mentioned). Hereafter, we only used data from the station with the longest record (Davis).
We also use data from automatic weather station (AWS) LGB59 located at the slope of the
Antarctic ice sheet inland from Progress station (Fig. 1), available for the period from 1994 to
1999, as well as surface air temperature data from Casey and Mawson.

2.4 Climatic indices of Southern Hemisphere
In order to investigate possible relationships between PEL climate multi-decadal variations and
large-scale modes of variability, we use data on the indices of the Antarctic Oscillation (AAO),
the Interdecadal Pacific Oscillation (IPO) and the Indian Ocean Dipole (IOD).
AAO index, also known as SAM (Southern Annular Mode), is defined as a mean latitudinal
difference of sea level pressure at 40 ºS and 65 ºS, and is considered as a prevailing mode of
Atmospheric circulation in the Southern Hemisphere representing about 35% of the extratropical
SH climate variability (Marshall, 2003). The monthly AAO index is available from NOAA:
http://www.cpc.ncep.noaa.gov/products/precip/CWlink/daily_ao_index/aao/monthly.aao.index.b
79.current.ascii.table (since 1979) and British Antarctic Survey
(http://www.antarctica.ac.uk/met/gjma/sam.html) since 1957, although data for the 1957-1978
period is considered to be less robust.
IPO is defined as a sea surface temperature (SST) anomaly over the Pacific Ocean. The positive
phase of IPO is characterized by relatively warm central and eastern tropical Pacific, and
relatively cold north-western and south-western Pacific (Henley et al., 2015;Dong and Dai,
2015). IPO index is closely related to PDO (Pacific Decadal Oscillation), but PDO better
characterizes Northern Pacific, while IPO is better applicable to the whole Pacific region. We
use IPO data because in the previous study we found a teleconnection between the climate
variability in the central Antarctic and tropical Pacific (Ekaykin et al., 2014).
The data on IPO index since 1870 is available here:
http://www.esrl.noaa.gov/psd/data/timeseries/IPOTPI/
IOD is characterized by Dipole Mode Index (DMI) that is defined as the SST gradient between
the western equatorial Indian Ocean (50 ºE - 70 ºE and 10 ºS - 10 ºN) and the south eastern
equatorial Indian Ocean (90 ºE - 110 ºE and 10 ºS - 0 ºN). Thus, IOD is an analogue of SOI
(Southern Oscillation Index), but for Indian Ocean. The data on DMI index since 1870 could be
found at:
http://www.jamstec.go.jp/frsgc/research/d1/iod/iod/dipole_mode_index.html.

**3 Results and discussion**
3.1 Surface air temperature variability in the Princess Elisabeth Land during the period of
instrumental observations (1958-2015)
Here, we first consider the variability of surface air temperature recorded at the meteorological
stations in Princess Elisabeth Land, to assess whether the studied sector is characterized by
uniform climate variability, and to provide a reference regional temperature record for
comparison with the δD stacked record.
Correlation coefficients between annual mean surface air temperature data at Vostok, Mirny and
Davis vary between 0.6 and 0.9 (Table 2). Correlation coefficients between Automatic Weather
Station LGB59 (located between Davis and Vostok, Fig. 1) and these 3 stations vary between
0.86 and 0.96. Despite the short record at LGB59, they are also significant at 95% confidence
level. These results demonstrate that the region encompassed between these 3 stations has
experienced similar climatic variability. This is further confirmed by a cluster analysis of surface
air temperature data from 12 Antarctic stations (see Supplementary Figure S1), showing that
Vostok, Mirny, Casey, Mawson and Davis data form a single cluster in terms of climatic
variability.
Interestingly, the correlation coefficient between Mirny and Vostok data is significantly weaker
in 1958-1976 (R=0.53) than in 1976-2015 (R=0.74). This suggests that, before the so-called
"1976 climate shift" (Giese et al., 2002) Vostok experienced a higher influence from the Pacific
sector of the Southern Ocean (Ekaykin et al., 2014) not encompassed at Mirny. Indeed, the
correlation coefficient between temperature data from Vostok and Mc Murdo Station (located in
the Pacific sector) was higher before the 1976 shift (R=0.46) than after 1976 (R=0.35).
During the whole period of instrumental observations, the strongest relationships observed for
temperature at Vostok were with temperature data at Mirny and Mawson coastal stations from
the Indian Ocean sector, and more precisely the sector between Davis Sea and Cooperation Sea.
As a result, Figure 4a shows the average temperature anomaly from Vostok, Mirny and Davis
stations. Hereafter, we use this stacked temperature record as an estimate of the temperature
anomaly for the whole PEL sector.
We now compare the low frequency variations in these various temperature records, using the
27-year low pass filter (Figure S2). Both Vostok and Mirny demonstrate a qasi-periodical
variability with a period of about 30 years, maxima in the late 1970s and the late 2000s, and
demonstrate a very high similarity at low frequency. While Davis data have the same periodicity,
their maxima are shifted to the early 1970s and early 2000s. If we consider other Antarctic
stations, we see a complex behavior of air temperature in different sectors of Antarctica: most
stations also show a 30-year cycle, but with a significant phase shift relative to PEL region.
In the Indian Ocean sector, temperature peaks appear more and more delayed when moving from
west to east. For example, the first maximum occurred late in the 1960s at Mawson, early in the
1970s at Davis, in the second half of the 1970s at Mirny, and late in the 1970s at Casey. This
feature may reflect a low-frequency component of the Antarctic Circumpolar Wave (Carril and
Navarra, 2001).
With respect to multi-decadal trends, contrasted patterns emerge: some stations (Esperanza,
Novolazarevskaya, Davis, Vostok, Mirny, McMurdo) display warming trend, while a cooling
trend emerges at Halley or Dumont d'Hurville (Figure S2).
This comparison of instrumental temperature records highlights different patterns of multi-
decadal variability across different sectors of Antarctica, which is important for interpreting
paleoclimate records, and for combining various proxy records for temperature reconstructions
(Jones et al., in press). Our analysis nevertheless demonstrates coherency within Princess
Elisabeth Land, where we will use the stacked temperature record from Vostok, Mirny and Davis
as a reference regional signal (hereafter named PEL temperature anomaly) for calibration of $\delta D$
records.

3.2 Relationships between Princess Elisabeth Land instrumental temperature records and
Southern Hemisphere modes of variability
Here, we compare the PEL temperature anomaly with indices that characterize climatic
variability in the Southern Hemisphere. First, as expected, a very strong negative relationship
with the AAO index (r = -0.68) is observed in 1979-2015 (Fig. 4b). The Antarctic Oscillations is
the predominant mode of climatic variability in Antarctica: a strong AAO index reflects a larger
pressure gradient between low and high latitudes, associated with a more zonal circulation
around Antarctica, and colder conditions in East Antarctica. We note that no correlation between
PEL and AAO is identified prior to 1979, which could be an artifact due to poor estimate of
AAO before 1979, when few instrumental records are assimilated in atmospheric reanalyses.
The correlation coefficient of PEL temperature anomaly with the IPO index is weak (Fig. 4c),
but the residuals of the PEL temperature regression with AAO are negatively correlated with
IPO index (r = -0.47).
A multiple linear regression approach leads to the conclusion that combined variations in AAO
and IPO explain 59% of the temperature variance, at the inter-annual scale. While such tele-
connection between Pacific and central Antarctic climate had previously been reported from
Vostok data (Ekaykin et al., 2014), the underlying mechanism is not known. Finally, no
significant correlation was identified between PEL temperature and the IOD index (Fig. 4d).
However, different results emerge when considering the low-pass filtered time series. At multi-
decadal time scales, a strong positive correlation (r = 0.8, significant with a 0,06 confidence
level) relates PEL temperature and the AAO (Fig. 4a and 4b), and a very strong positive
correlation appears between PEL temperature and the IOD index (r = 0.93, p < 0,05). We suggest
that the Indian Ocean Dipole affects the Antarctic climate through a modulation of cyclonic
activity. This is indirectly confirmed by a negative correlation (r = -0.56) between the IOD index
and the pressure anomaly at Mirny and Davis (not shown). The positive relationship between
AAO and temperature in the low frequency band could then be an "induced correlation" caused
by a very strong positive correlation between AAO and IOD (r = 0.8-0.9) at these time-scales.

3.3 Climatic variability in Princess Elisabeth Land over the last 350 years
The stacked δD record (built from low-pass filtered individual records)  is now compared with
the filtered PEL temperature composite (Fig. 4a). We observe a positive correlation with r =
0.66. Although the length of the series is 52 years, the number of degree of freedoms is only 4,
due to the 27-year filtering. The uncertainty of the correlation is ±0,4, so it is statistically
insignificant (p = 0,17).
This invokes a discussion of the factors that may disturb the correlation between the local air
temperature and the stable water isotopic composition of precipitation in Antarctica (Jouzel et
al., 2003).
Firstly, isotopic composition of precipitation is not a function of local air temperature, but of the
temperature difference between the evaporation area and the condensation site, which defines the
degree of heavy water molecules distillation from an air mass. The study of the moisture origin
for this sector of Antarctica (Sodemann and Stohl, 2009) demonstrates that different parts of the
PEL differ in their moisture origin. Coastal areas receive moisture from higher latitudes (46-52°
S) and from more western longitudes (0-40° E) than inland areas (34-42° S and 40-90° E). It
means that even if our sector is climatically uniform, as was shown above, the temporal
variability of the precipitation isotopic content may differ in the different parts of the sector due
to varying moisture origin.
Secondly, we should define which temperature is actually recorded in the isotopic composition
of precipitation. For central Antarctica, where much (or most) of precipitation is "diamond dust"
from clear sky (Ekaykin, 2003), the effective condensation temperature is conventionally
considered equal to the temperature on the top of the inversion layer. But it is definitely not true
for the coastal areas, where most precipitation falls from clouds. Thus, the difference between
near-surface and condensation temperature may vary in space and time.
Thirdly, the precipitation seasonality is another factor that may change the relationship between
the air temperature and stable isotope content in precipitation. At Vostok the precipitation
amount is evenly distributed throughout the year (Ekaykin et al., 2003), so the snow isotopic
content corresponds well to the mean annual air temperature, but we don't have robust
information neither about the other parts of the PEL, nor about the seasonality changes in the
past.
Yet we believe that the main factor that affects the isotope-temperature relationship is the
"stratigraphic noise". Indeed, even when we study the ice cores obtained in a short distance one
from another (Ekaykin et al., 2014), the correlation between the individual isotopic records is
still small, despite the same climatic conditions.
This is why we argue that constructing the stacked isotopic record is an optimal way to reduce
the amount of noise in the series and to highlight the variability that is common for the whole
studied region, provided that the region is climatically uniform.
Despite the statistically insignificant correlation coefficient, we assume that the stacked $\delta D$
record is a proxy of surface air temperature in the PEL region (or, following Steig and others
(2013) a proxy that "covaries with atmospheric circulation in a manner similar to temperature").
Thus we estimate the calibration coefficient between these two parameters as a ratio of the
standard deviation of the $\delta D$ composite record to the standard deviation of the PEL low-pass
filtered temperature record, which allows us to assign a temperature scale to the isotopic record.
The apparent isotope-temperature gradient, obtained as a standard deviation of isotopic values
divided by standard deviation of temperature values, is 13,8±2,5 ‰ °C$^{-1}$ (the uncertainty is due
to different standard deviation of isotopic values in individual records). Such an approach
implicitly suggests a perfect correlation between the compared series. If we correct the apparent
slope by the observed correlation coefficient, 0.66, it becomes 9±5.4 ‰ °C$^{-1}$. The latter value is
still higher than the corresponding slopes observed in other regions of Antarctica (see a review in
(Stenni et al., 2016)), but corresponds nicely to an isotope-condensation temperature slope
predicted by simple isotope model (Salamatin et al., 2004). Actually, low apparent isotope-
temperature slopes obtained based on ice-core data may be due to significant amount of noise in
the isotopic records, while in our case we considerably removed noise by filtering and
constructing the stacked record.
The temperature reconstruction is displayed in Fig. 5b as a temperature anomaly relative to the
1980-2009 period. We also show the instrumentally obtained air temperature anomaly in Fig. 5b
on the same temperature scale.
Following (Ekaykin et al., 2014), who reported a closer relationship between Vostok isotopic
data and summer temperature than with annual mean temperature, we performed additional
analyses of relationships between our stacked isotope record and other temperature time series
(e.g. monthly or seasonal temperature anomalies), but this did not improve the isotope-
temperature correlation.
Despite discrepancies in the individual isotopic records (Fig. 3), common signal identified in the
stacked record lead to several conclusions about PEL climate variability over the past 350 years
During this time interval, regional surface air temperature shows a long-term increasing trend,
and an overall warming by about 1±0.6°C. Superimposed on this multi-centennial trend, quasi-
periodical variability occurs with periods of 30-40 and about 60 years. A colder period is
identified in 1750-1860 - i.e., approximately at the same time interval as the "Little Ice Age"
reported in the other regions (PAGES 2k network, 2013).
A remarkable cold phase is observed during the 1840s, during which PEL temperature could fall
1.2±0.7 °C below present-day (defined as the average value of the last 30 years). As seen in Fig.
3, this event is a robust feature, observed in all 4 individual records available for this time
interval. This minimum was also identified in an Antarctic temperature stack record (Schneider
et al., 2006) – see Fig. 5d, as well as in an ice core drilled in the Ross Sea sector (Rhodes et al.,
2012) and in the isotope record from Ferrigno (coastal Ellsworth Land) (Thomas et al., 2013).
Further studies are needed to understand whether such remarkable cold conditions arise from
internal variability or are driven by the response of regional climate to an external perturbation.
A possible candidate could be a response to volcanic forcing (Sigl et al., 2015). A moderate
event is associated with the eruption of Cosigüina in 1835. According to the inventory of
volcanic events recorded in the Vostok firn cores (Osipov et al., 2014), there was an eruption of
an unknown volcano in 1840; however, the amount of deposited sulfate was about 15% of that of
Tambora, so it is not expected to have a major effect on climate system. So far, the influence of
volcanic forcing on Antarctic climate, and the response time remains poorly known. By contrast,
recent studies have stressed the delayed response of the North Atlantic Oscillation (Ortega et al.,
2015) to major volcanic eruptions, as well as their role as pace-makers of bidecadal variability in
the North Atlantic (Swingedouw et al., 2015).
The period before 1700 is probably the coldest part of the record, but this is not a robust result as
the 2 records spanning this time interval show somewhat different behaviors (Fig. 3). However,
another stack of 5 East Antarctic cores from PAGES 2k (Fig. 5e) also highlights that the 1690s
could have been the coldest decade of the last 350 years.
We also compare the PEL2016 record with other Antarctic temperature reconstructions.
(Schneider et al., 2006) used high-resolution isotopic records from 5 Antarctic sites (a stack of
Law Dome records, Siple Station, a stack of Dronning Maud Land records, and two ITASE sites
from West Antarctica). Although this record is not significantly correlated with PEL2016 (r =
0.36), we note some common features in both records (warming in the 1820s and 1890s, cold
events in the 1840s and 1900s, etc.).
We also investigated the similarities between PEL2016 and the filtered stack normalized isotopic
East Antarctic record based on 5 East Antarctic ice cores (Fig. 5e; data are available in
Supplementary materials of (PAGES_2k_network, 2013)). The correlation with PEL2016 is
weak (r = 0.13) and insignificant, and so is the correlation with the stack from Schneider et al
(2006) (r = 0.36).
The main difference between our PEL2016 record and the other isotopic stacked records for the
whole Antarctica (Fig. 5d) and for East Antarctica (Fig. 5e) appears for long-term trends, with a
long-term increase in PEL2016 but no similar feature in the other reconstructions. We suggest
that contrasted regional long-term trends may disappear in continental-scale reconstructions (see
Fig. S2).
Finally, we compare our PEL2016 record with an IOD time-series since 1870, also processed
with a low-pass filter. The strong correlation coefficient (r=0.79) confirms the tight relationship
between multi-decadal variations in surface air temperature in this sector of Antarctica and IOD.
The Indian Dipole Ocean oscillation appears as the predominant climatic mode affecting multi-
decadal climate variability in this part of East Antarctica. While the exact mechanisms
underlying this relationship are not known, the IOD is expected to affect the inland Antarctic
climate by modulating the cyclonic activity that brings heat and moisture to Antarctic continent.

3.4 Snow accumulation rate variability
We now investigate the low-pass filtered values of snow accumulation rate, available at sites
"105 km", "200 km" and Vostok (the latter is a stack curve from 3 deep snow pits), normalized
over the period from 1952 to 1981 (Fig. 6). All of them exhibit a negative trend, more prominent
for "200 km" series. This result contradicts the stacked Antarctic snow accumulation rate record
(Frezzotti et al., 2013) showing an overall increase of the accumulation rate during the last 200
years. Our finding is also not supported by the accurate assessment of average accumulation rate
change between successive reference horizons at Vostok, showing a slight but significant
increase of snow accumulation rate since 1816 (Ekaykin et al., 2004). Our results moreover
stress the fact that, during the last centuries, opposite long-term trends may have occurred in
temperature and accumulation. This is counter-intuitive with respect to atmospheric
thermodynamics and to the expected co-variation of heat and moisture advection towards inland
Antarctica. Similar divergence of the centennial trends of snow isotopic composition and
accumulation rate was observed by Divine et al. (2009) at the coastal sites of Dronning Maud
Land, but not at the inland sites (Altnau et al., 2015).
Processes other than snowfall deposition may however affect the ice core records. In the vicinity
of "105 km", large "transversal" snow dunes have recently been evidenced (Vladimirova and
Ekaykin, 2014). Such features may lead to a strong non-climatic variability in the snow
accumulation rate in a given point, due to dune propagation effects. Blowing snow events may
also have a significant influence on mass balance in the coastal zone of Antarctica (Scarchilli et
al., 2010), potentially introducing additional post-deposition noise.
As a result, we are not confident that the datasets reported in Figure 6 can be interpreted in terms
of climate (snowfall) variations, and further work is needed to decipher the large-scale climate
effect (snowfall deposition) from the non-climatic effects potentially associated with post-
deposition (wind erosion, dune propagation etc).

## 4 Conclusion

In this paper, we presented an analysis of the recent variability in snow isotopic composition (δD) data from 6 snow pits and ice cores recovered in the region of Princess Elisabeth Land (PEL), East Antarctica.

To interpret this data, we have investigated the present-day mean annual surface air temperature variability using the instrumental temperature measurements at stations Mirny, Davis and Vostok located at the margins of the studied sector. It was shown that inter-annual climatic variability strongly covariates at these three stations. Cluster analysis demonstrated coherent variations for these stations, together with the nearby stations of Casey and Mawson. However, we have stressed phase shifts between multi-decadal temperature variations along the coastal stations: temperature maxima and minima at Vostok and Mirny are delayed by a few years compared to those at Davis. At a broader geographical scale, temperature records from different sectors of Antarctica exhibit different climatic variability at decadal scale in terms of periodicities, phasing and trends.

We then compared recent temperature variability in the PEL region with indices of Southern Hemisphere modes of variability, and highlight the importance of the Annular Antarctic Oscillation and the Interdecadal Pacific Oscillation that in total explain 59% of the temperature variance in this Antarctic region. At the multi-decadal time-scale, however, temperature variations appear most closely related with the Indian Ocean Dipole mode, suggested to modulate the cyclonic activity bringing heat and moisture to Princess Elisabeth Land.

Given limitations of ice core data for inter-annual variations, we have processed our isotopic time-series with a low-pass filter to cut off variability expressed at timescales <27 years. Both common features and significant discrepancies emerge from individual filtered time-series. These differences may arise from true differences in regional climate variations, and/or by non-climatic noise.

In order to improve the signal-to-noise ratio, we constructed a stacked isotopic record for the Princess Elisabeth Land based on data from all 6 sites. We then used the linear regression between this record and instrumentally obtained air temperature record in order to convert the isotopic composition scale into air temperature scale. The apparent isotope-temperature slope is $9\pm5,4$ ‰ °C$^{-1}$.

The newly obtained temperature reconstruction covers the period from 1654 to 2009. During this period, temperature appears to have gradually increased by about $1\pm0.6$ °C, from a relatively

cold period observed from the mid-17[th] to mid-19[th] centuries. The coldest decade is identified in the 1840s, a feature common to several Antarctic isotopic composite signals. By contrast, long-term temperature trends were not identified previously in pan-Antarctic stacked records, possibly due to averaging effects of different regional trends. We found a weak positive correlation of our temperature reconstruction with reconstructions previously obtained for the whole Antarctic continent and/or East Antarctica. A poor correlation between different Antarctic temperature records based on ice core data from different (but partly overlapping) regions requires further improvements of the ice core-based climate reconstructions.

Finally, our PEL record appears closely related to the low-frequency component of the Indian Ocean Dipole mode.

The three accumulation time series depict decreasing long-term trends and large inter-site differences. Further investigations of non-climatic drivers (including wind erosion and dune effects) are needed prior to confident climatic interpretation.

Our time-series is provided as supplementary information to this manuscript. Understanding the cause for the reconstructed changes will require to compare the PEL record with other regional Antarctic records, expanding the work of Jones et al (in press), and combining simulations and reconstructions in order to better understand the mechanisms of regional climate multi-decadal to centennial variations, and to explore the potential response of Antarctic climate to external forcing factors (e.g. volcanic eruptions).

This study finally stresses the importance of obtaining a dense network of highly resolved ice core records in order to document the complexity of spatio-temporal variations in Antarctic climate, a key focus of the Antarctic 2k project (http://www.pages-igbp.org/ini/wg/antarctica2k/intro).

**Acknowledgement**

This work is a contribution to PAGES and IPICS "Antarctica 2k" project. We are grateful to all the field technicians of Russian Antarctic Expedition (RAE) and drillers from St. Petersburg Mining University for providing us with the high-quality ice cores. We thank RAE for logistical support of our works in Antarctica. The Russian-French collaboration in the field of ice cores and paleoclimate studies is carried out in the frames of International Associated Laboratory "Vostok". We thank the CERL's staff for the isotopic analyses. The chemical analyses of the samples were performed at Irkutsk's Limnological Institute of RAS in frames of Russian

470    Foundation for Basic Research grant 15-55-16001. One of the authors (VMD) was supported by

471    Agence Nationale de la Recherche in France, grant ANR-14-CE01-0001.

472    This study was completed with a financial support from Russian Science Foundation, grant 14-

473    27-00030.

474

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

Table 1. Information on sites where individual time-series were obtained

| Site / series | Coordinates | | Alt., m above s.l. | Time interval, years AD | Acc. rate, mm w.e. | Sample resolution, cm / Number of samples per year | δD measurements | Accumulat record ava |
|---|---|---|---|---|---|---|---|---|
| | Lat., °S | Long., °E | | | | | | |
| 105 km | 67.433 | 93.383 | 1407 | 1757-1987 | 310 | 5 / 15 | LSCE, mass spectrometry; CERL, laser spectroscopy | Yes |
| 400 km | 69.95 | 95.617 | 2777 | 1254-1987 | 170 | 100 / 0.4 | LSCE, mass spectrometry | No |
| VRS 2013 stack (Vostok) | 78.467 | 106.84 | 3490 | 1654-2010 | 21 | 1-7 / 1-6 | LSCE, mass spectrometry; CERL, laser spectroscopy | Yes |
| NVFL-1 | 77.11 | 95.072 | 3775 | 1711-1944 | 31 | 10 / 1 | CERL, laser spectroscopy | No |
| NVFL-3 | 76.405 | 102.167 | 3528 | 1978-2009 | 34 | 10 / 1 | CERL, laser spectroscopy | No |
| PV-10 | 72.805 | 79.934 | 2800 | 1976-2009 | 103 | 2 / 12 | CERL, laser | No |
| 200 km | 68.25 | 94.083 | 1990 | 1640-1987 | 271 | NA | no | Yes |

NA = not applicable


Table 2. Correlation matrix between individual surface air temperature records from meteorological
stations in the Princess Elisabeth Land.

|        | Casey | Mirny | Davis | Mowson | Vostok |
|--------|-------|-------|-------|--------|--------|
| Casey  | 1     | 0.82  | 0.60  | 0.53   | 0.54   |
| Mirny  |       | 1     | 0.86  | 0.77   | 0.67   |
| Davis  |       |       | 1     | 0.86   | 0.58   |
| Mowson |       |       |       | 1      | 0.62   |
| Vostok |       |       |       |        | 1      |

All the correlation coefficients are statistically significant with 95 % confidence level.


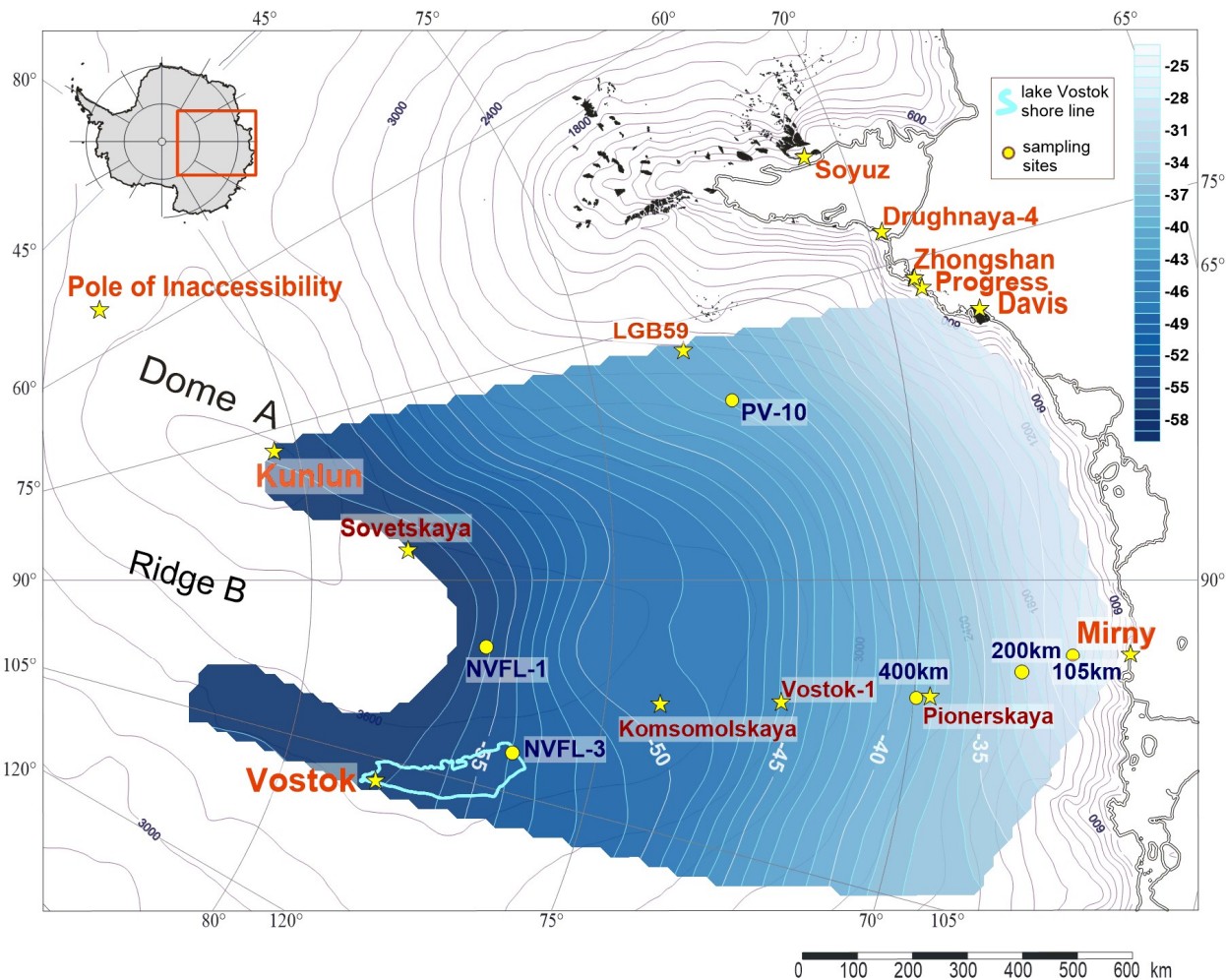


Figure 1. The Princess Elisabeth Land sector of East Antarctica. Blue iso-contours display the spatial pattern of surface snow $\delta^{18}$O (Vladimirova et al., in preparation). The light blue contour shows the shoreline of subglacial Lake Vostok. Yellow dots mark the location of individual records used here. Stars depict the location of former or present research stations.



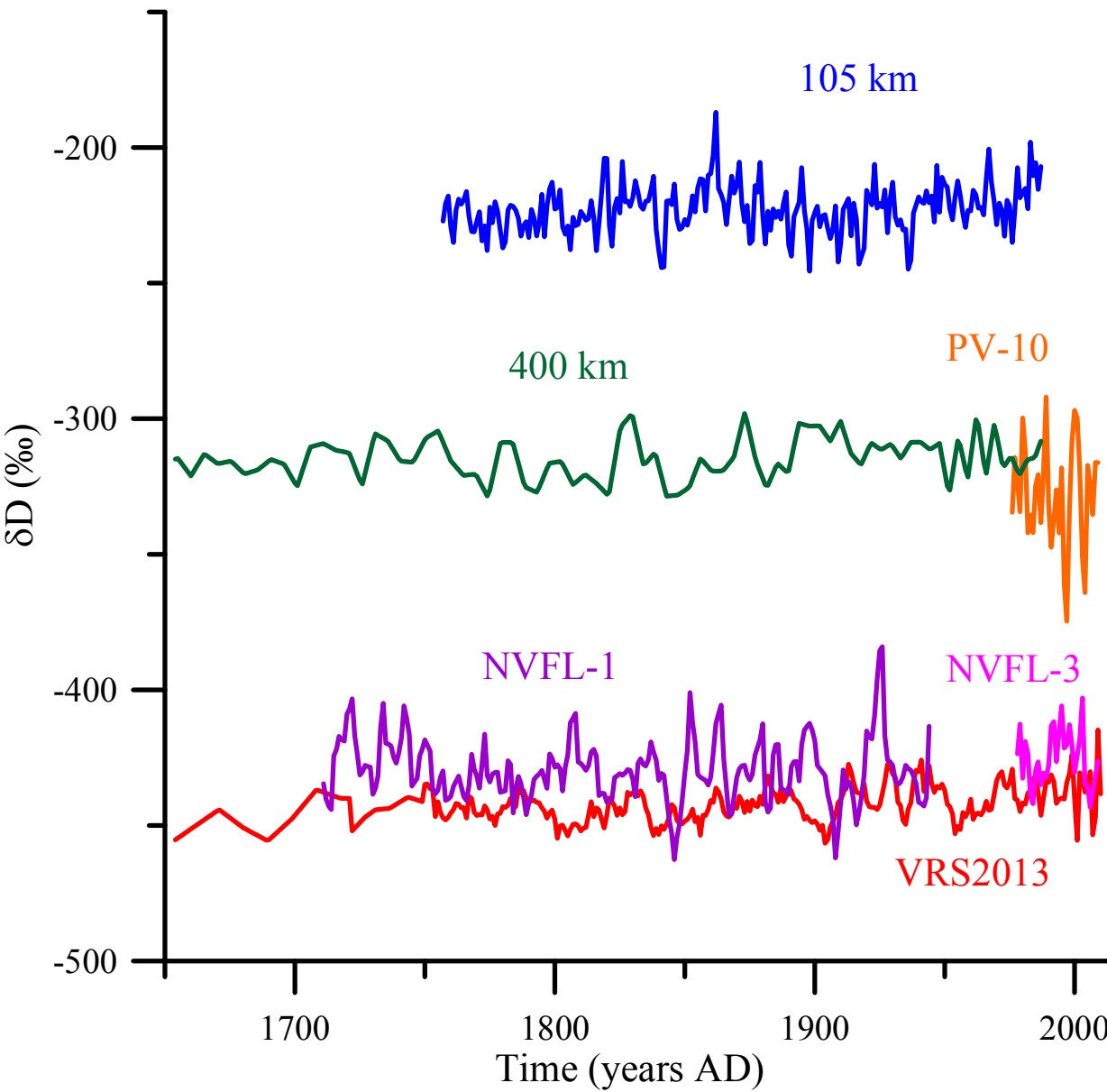


Figure 2. δD records from 6 individual series used in this study.


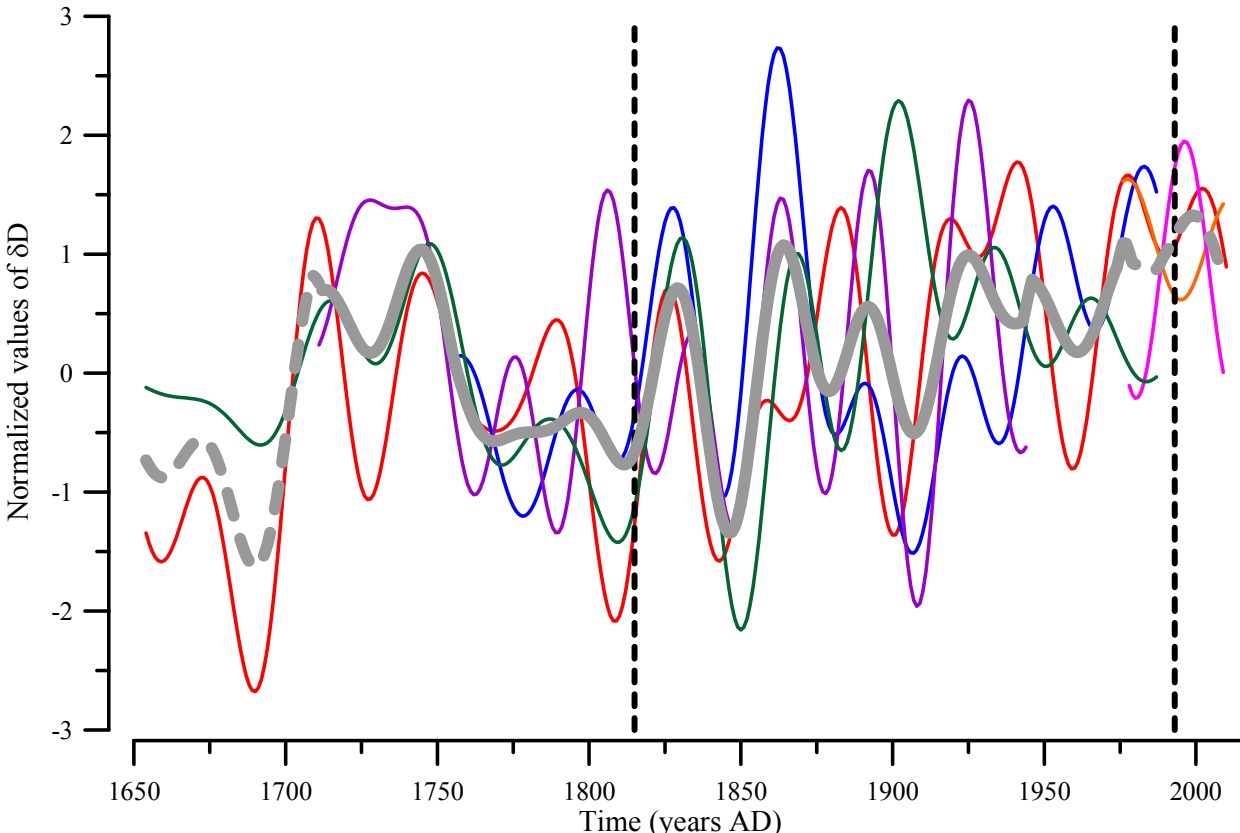


Figure 3. Normalized and low-pass filtered individual records (with a cut-off for variations on timescales shorter than 27 years)-, displayed using the same colors as in Figure 2.

The thick grey line is the stacked record (PEL2016). The dashed grey lines show the less robust marginal parts of the stack.

Vertical dashed lines mark reference horizons that contain the debris of Tambora (1815) and Pinatubo (1991) volcanic eruptions, respectively deposited until 1816 and 1993 in Antarctica.

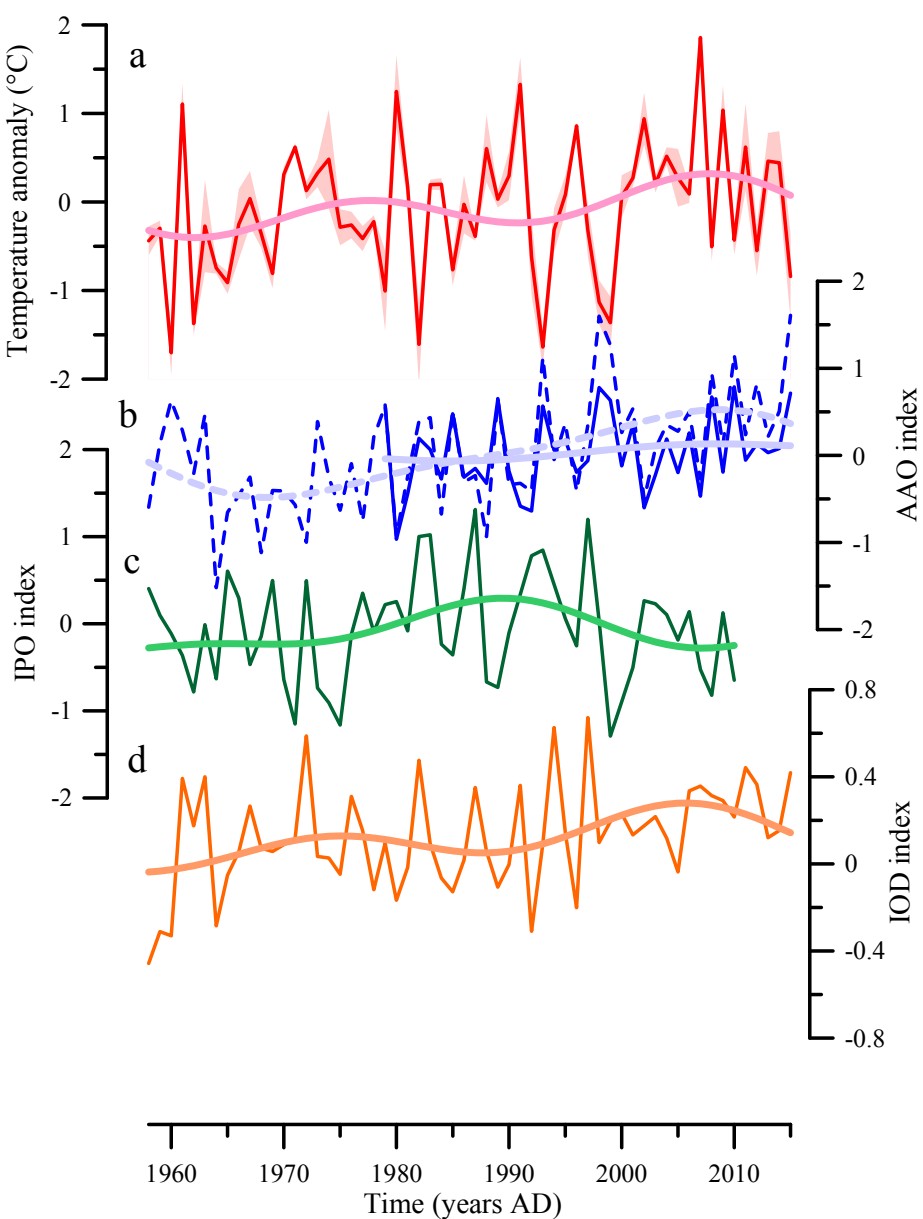

613

Figure 4. Climatic variability in the Southern Hemisphere in 1958-2015.

a – Composite temperature anomaly in the Princess Elisabeth Land (based on records from Mirny, Davis and Vostok). The red shading displays ± 1 standard error of mean.

b – Antarctic Oscillation Index from NOAA (solid line), and BAS (dashed line). See text for details.

c – Interdecadal Pacific Oscillation Index.

d – Indian Ocean Dipole Index.

Thick lines are low-pass filtered (with a cut-off for variations on timescales shorter than <27 years).

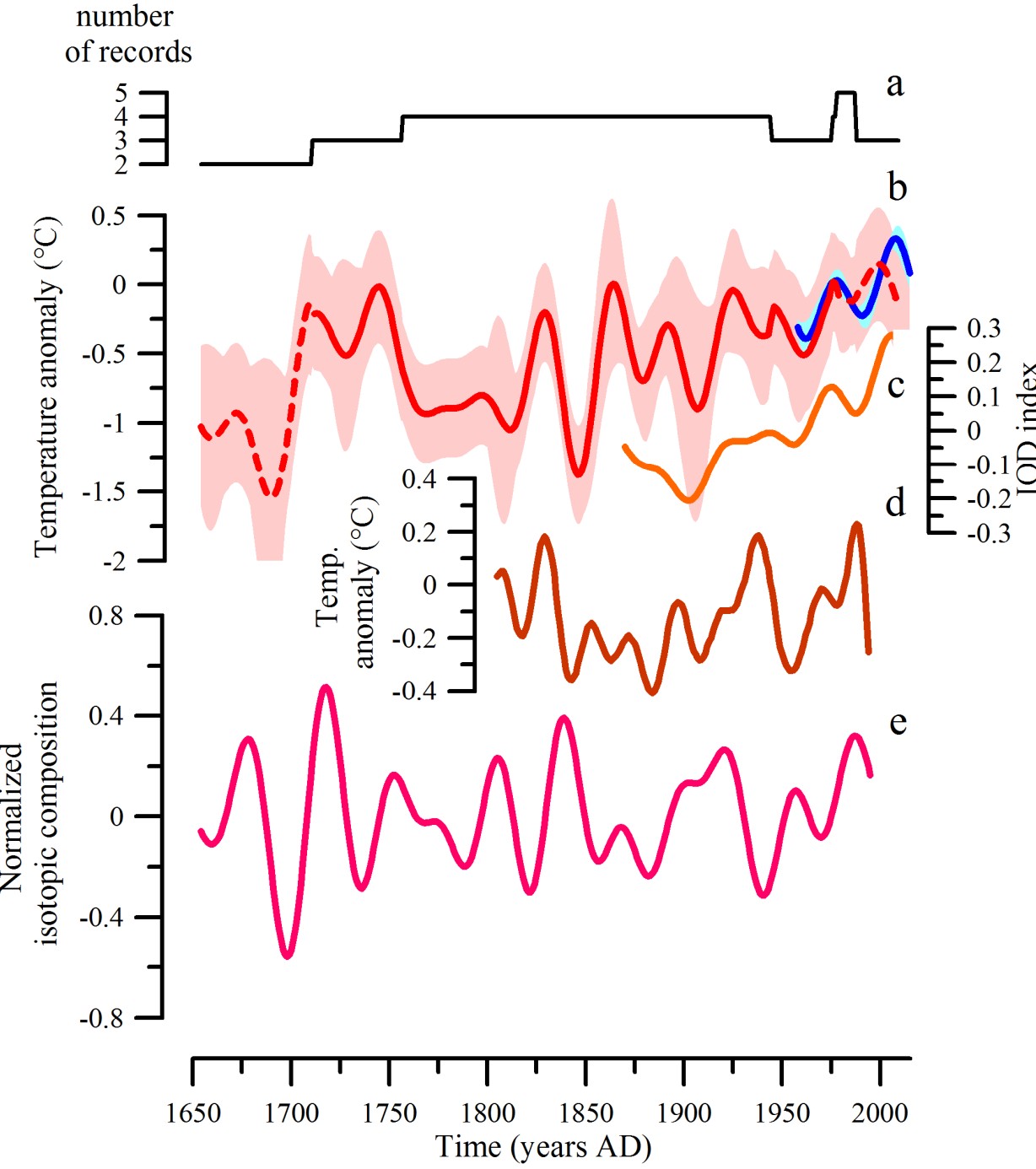


Figure 5. Antarctic climatic variability over the past 350 years.

a – Number of individual records in the stacked isotopic record;
b – Temperature anomaly relative to 1980-2009, based on Princess Elisabeth Land
meteorological records (blue) and reconstructed from the stacked isotopic record (PEL2016 –
red). Shading is ±1 standard error of mean. Dashed lines denote less robust marginal parts of the
PEL2016 record.
c – Low-pass filtered values of the IOD index.
d – Antarctic temperature anomaly from (Schneider et al., 2006).
e – Normalized and low-pass filtered stacked isotopic record for East Antarctica (data from
(PAGES_2k_network, 2013)).

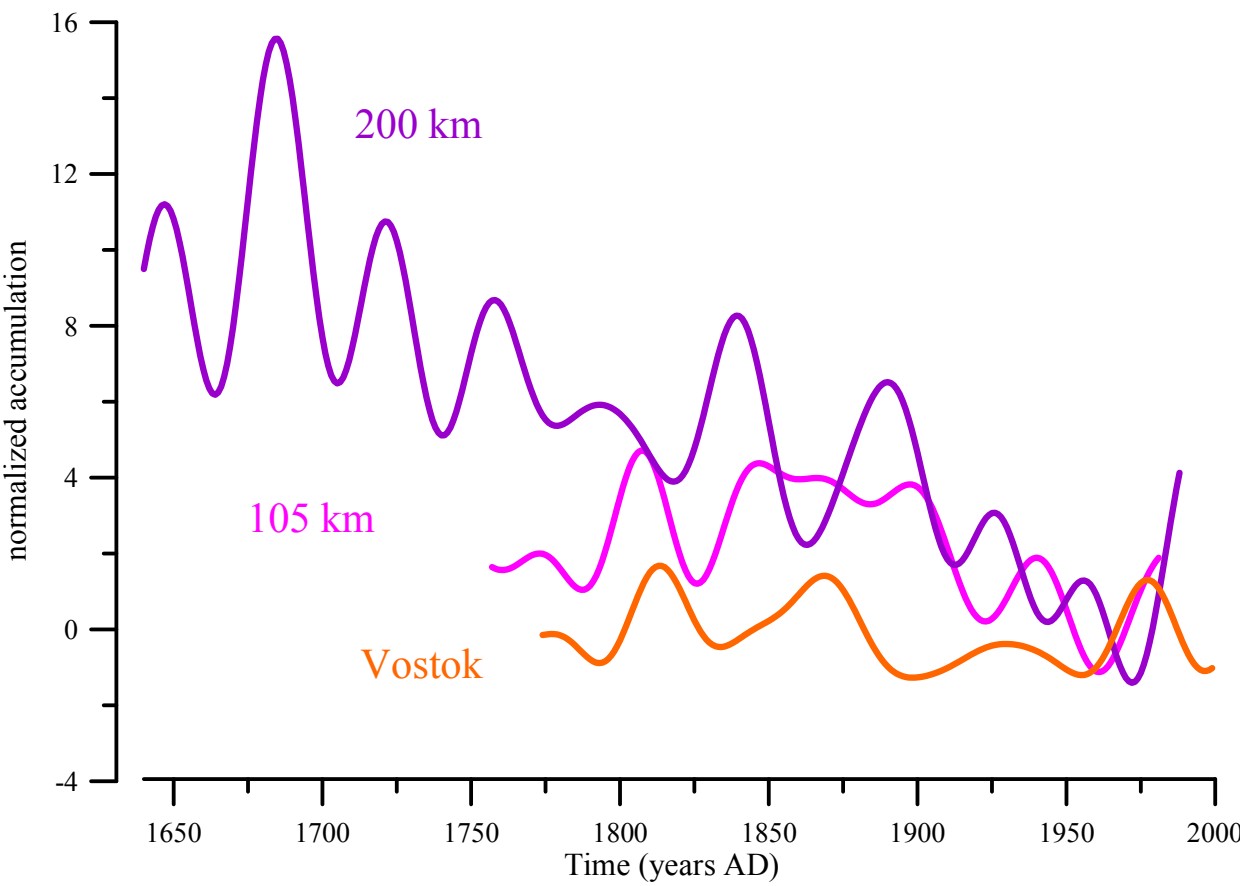


Figure 6. Normalized (relative to period 1952-1981) and low-pass filtered records of snow accumulation rate at sites "200 km" (purple), "105 km" (magenta) and Vostok (orange).

