# Peer review of "Climatic variability in Princess Elizabeth Land (East Antarctica) over the last"

_Climate of the Past, 2016_

## Referee Comment (RC1) · D. Divine (Referee) · 18 Jul 2016

**Review of a manuscript for *Climate of the Past***

**Climatic variability in Princess Elizabeth Land (East Antarctica) over the last 350 years** by Ekaykin et al.

**Overall:**

In this manuscript the authors present an ice core based reconstruction of 350 years of temperature variability in Princess Elisabeth Land, Antarctica. The reconstruction is derived using scaling of the smoothed \deltaD stack of 6 ice cores from the plateau to the stacked instrumental annual mean temperature series from Vostok, Mirniy and Davis stations. The authors present evidence of a persistent positive trend in the reconstructed regional temperatures observed over the length of the stack together with the pronounced variations on bidecadal and multidecadal timescales. This work is a contribution to Antarctic Pages 2K initiative aimed at reconstructing and understanding regional climate variability over the last 2000 years.

In general the paper is clearly written and results are well presented; it fits the scope of the special issue very well and deserves to be published after some modifications to the content have been made. I have some relatively moderate methodological comments that the authors are encouraged to address/answer before the paper can be accepted.

As a non-native English speaker I can not comment much on the language quality. However, my impression was that some style improvement/language check by a native speaker would be highly desirable to improve the manuscript readability and eliminate some language flaws.

**Major comment**

1) My first major comment concerns the method the authors used to estimate the isotope to temperature gradient and its STD on the smoothed data. More specifically, it is not demonstrated that a reduced number of degrees of freedom (DOF) in the data due to smoothing is taken into account. The same applies to significance of the correlation coefficients reported for the smoothed series. For a 27-year low pass filtered instrumental series of a length of about 60 years one have to expect about 5 independent data points only, implying that a simple sample variance (or STD) of the slope presented in the manuscript is a biased estimator of an underestimated true variance. For a very simplified case of AR(1) model of serial correlation in the data, taking the effect of autocorrelation into account to estimate the confidence intervals (CI) on the slope estimate was summarized in Nychka et al., 2000 (available from http://citeseerx.ist.psu.edu/viewdoc/download;jsessionid=A30C325B3A1E36EAB30B126EF74F974E?doi=10.1.1.33.6828&rep=rep1&type=pdf ). To reassess the significance of correlation coefficients simple adjustment for a number of independent samples in the t-distribution quantile can be applied as a simplistic remedy of the problem.

2) Some discussion on precipitation types/seasonality, and moisture origin that can be different for the coastal and inland locations in the study area would be highly relevant in the context of the observed discrepancies between the core series and the instrumental data.

**Other comments**

Page 1 last line: "the only source of climatic data". Please use "primary" instead; there are alternative though sparse sources of instrumental data such as earlier expeditions to Antarctica, observations from ships logbooks etc.

Page 2 Line 5: "…moreover unevenly distributed…", ".reflecting heterogeneous efforts…", "still remain white spots". Awkward sentences, please check the language.

Page 2, Line 15: "Classically" can be omitted.

Page 2, Line 29: "…down to a 150 m depth…"

Page 2: "Individual records" can be modified to "ice core data"

Pages 2-3, Section 2.1: Q. on ice core dating. Did the authors use, wherever possible, counting the seasonal peaks in d18O to establish and/or support their core chronologies?

Page 3, Line 16: The age uncertainty associated with the Nye model alone can also be estimated directly from the Nye formula, please see Divine et al., 2011 (*Polar Research*, 30, 7379, DOI: 10.3402/polar.v30i0.7379, on page 3) for details.

Page 3, Line 27: "…values were reduced in terms of mean and STD…". Awkward sentence, better to refer to the procedure as a "mean and variance adjustment" or a "variance scaling" (see e.g. Esper etal., 2005, GRL 32, L07711, doi: 10.1029/2004GL021236.

Page 3, Line 27: "…to avoid an artificial dominating…", please check the language.

Page 3, Line 29: "…to cut off the variability with periodicities lower than 27 years…". Use "shorter" rather than "lower". Please provide some more detail on the filtering procedure you have actually used.

Page 4, Line 2: "…due to a very low SNR…" …and non-temperature effects on isotopes in precipitation including post-depositional alterations.

Page 4, Line 4: "…despite (some) common features… "

Page 4, Line 8: "…observed discrepancies do not arise from chronological uncertainties alone…"

Page 4, Line 9: "…significant level of noise event in the filtered series". …and other than the ambient temperature -related controls on the isotopic composition of precipitation.

Page 4 Section 2.3. Subsection title can be changed to "Instrumental temperature data"

Page 4 Line 17. "The data are available from…". Please mention explicitly that the annual means were constructed from the monthly means.

Page 5 Line 2: "…considered as a prevailing mode of atmospheric circulation in the SH representing about 35% of the extratropical SH climate variability".

Page 5 Line 2: "The monthly AAO index is available from…"

Page 5 Line 22: "…to assess whether uniform climate variability pattern is monitored…". Awkward sentence, consider revision.

Page 5 Line 26. High correlation coefficient reported for AWS LGB59, is it based on 5 annual values only or the authors used the monthly means for this particular case? If the latter is correct did the authors subtract the annual cycle from the data?

Page 5 Line 27: "…that the region encompasses between these 3 stations…". Please check the language and consider revision.

Page 5 Line 28: Just a comment: principal component analysis commonly used in climate sciences, could be considered a reasonable alternative to a cluster analysis.

Page 6 Line 14: "…have a 30-year periodicity…". Due to a shortness of the data being analyzed, referring to a "quasi-periodic variability" would be more appropriate. Mind also the edge effects of any filtering procedure that in the zone of influence equal to a filter length at a specified timescale.

Page 7 Line 5: "…reflects a larger pressure gradient…"

Page 7 Line 15: please see my major comment 1.

Page 8 Lines 3-5: since the presented slope estimate is based on the low-pass filtered series, a decreased number of DOF needs be taken into account. The STD on the estimated slope is presently underestimated and should be corrected; some more details on the method the uncertainty of the slope was calculated should be provided too.

Page 9 Line 23: "…the IOD is expected to affect the inland Antarctic climate…" can the authors provide any relevant reference pointing to a link between IOD and cyclonic activity in the coastal Antarctica?

Page 10, Line 4: A similar divergence in the longer term trends in d18O and accumulation was also observed for the coastal DML (see Divine et al., 2009, JGR,114, D11112, doi:10.1029/2008JD010475 ) but not on the plateau where both d18O and SMB showed positive trends (Altnau et al., 2015).

Page 10, Line 27: "…suggested to modulate…"

Page 11, Line 8: please provide STD on the estimated slope.

Page 11, Line28: "field technicians" or "field engineers" would be a more appropriate term.

Page 12, Line 1. "…in the framework…" , please indicate what abbreviation "LIA" stands for.

Figures

Figure 5: please use different colors for 5b. The lines are difficult to discriminate with the presently used color palette. Correct the uncertainty interval on the reconstruction by adjusting for the number of DOFs.

---

## Referee Comment (RC2) · E. Thomas (Referee) · 28 Jul 2016

The paper presents a new climate reconstruction for Princess Elizabeth land, based on the stable isotope records from a stack of six ice cores. The data makes an important contribution to the PAGES Antarctica 2k initiative, providing centennial scale records in a data sparse region. The paper is generally well written and should be published following a few minor corrections.

General comments:

Page 2 Line 1- some might consider borehole or historical records. Perhaps reword to "primary" or "a valuable source"

Line 13 – word missing "we find evidence of . . .", or "we observe a . . ."

Page 4, Line 4 – suggest remove "clear"

Page 4, Line 8 – suggest replace "only" with "solely"

Page 4, Line 21 – are all the correlations done on de-trended data?

Page 5, Line 1 - I know you are choosing to use the term AAO but perhaps an "also known as the SAM" would be helpful. The structuring of this paragraph could be improved. Consider using "the AAO index is available from NOAA (include web link in brackets) and the British Antarctic Survey"

Page 5, Line 11 – not sure if this was a mistake but should PDO be IPO? You are justifying the use of IPO because of a previous teleconnection with IPO?

Page 5, Line 15 – reference to SOI that is not defined in the text

Page 5, Line 22 – suggest changing "monitored" to "observed"

Page 5, results and discussion

Somewhere in this section is would be good to include reference to the moisture source regions or airmass transport routes. Has any backtracjectory work been done in this region that you could reference? This might aid the discussion about the differences between stations?

Page 8, Line 20 – can you add a short description of the little ice age? E.g. Cold period observed in northern hemisphere? I am a little nervous about defining LIA periods for Antarctic records. The pages 2k paper you cite states "There were no globally synchronous multi-decadal warm or cold intervals that define a worldwide Medieval Warm Period or Little Ice Age". Concluding that "a cold period is observed at approximately the same time interval as the little ice age reported in other regions" may be safer.

Page 8, Line 21 – Just for interest and comparison we also see a cold phase during the 1840s in the isotope record from Ferrigno (coastal Ellsworth Land). Might add evidence to it being a continental scale event. Thomas, E. R., T. J. Bracegirdle, J. Turner, and E.

W. Wolff (2013), A 308 year record of climate variability in West Antarctica, Geophys. Res. Lett., 40, doi:10.1002/2013GL057782

Page 9 - Snow accumulation variability. This section is lacking information on the thinning functions applied to the records. You mention the Nye model was used for the 400km core but nothing about the 105 and 200km records. Please just specify which thinning method was used in the text.

Page 10, Line 18 – suggest changing "has evidenced" for "demonstrates"

Page 11, Line 10 - suggest changing "evidenced" for "observed"

Table 1 – Suggest "this study" instead of "this work" For the sample resolution can you give an indicator of the number of samples per year? Or per decade for 400 km?

Figure 1 – Just a style issue but I found it hard to see the ice core locations on my screen. Consider changing the orange used.

---

## Short Comment (SC1) · 6 Sep 2016

I would like emphasize the importance of Dmitry Divine's comment that the effect of smoothing on the correlation and regression should be considered.

A large part of the conclusions hinges on the finding that there is a significant relationship between the ice-core stack and instrumental temperatures. However, this relationship is based on a 52 year overlap of 27 year lowpass filtered data.

Repeating the same exercise with random data (white independent noise, 27year lowpass, finite response filter, filter length 41yr, minimum norm endpoint constraint) suggests that the relationship of the ice-core stack and the instrumental data is not significant ($p>0.1$). The real data and some examples of random data (which are by definition unrelated) are shown in the attached figure.

[Figure]

The same also applies to the relationship with the PAGES stack ($\sim$350yr overlap, R=0.13, p>0.2) and Schneider stack ($\sim$200yr overlap, R = 0.36, p$\sim$0.1), only leaving the relationship to the IOD index significant, as long as the linear trend is not removed.

This is not just a statistical subtlety as the strength of the temperature to isotopic composition relationship in high resolution records derived from Antarctic low-accumulation regions is under debate. I still think that this is a very useful manuscript as it presents new records in a data-sparse region. However, if my assertions are confirmed, I would propose to tone down the temperature interpretation of the record (e.g. "1C warming over the last three centuries") and either avoid to provide a temperature calibration, or to provide proper uncertainty bounds.

―――――――――――――――――――――

**real data**

**Random realization 1**

**Random realization 2**

**Random realization 3**

**Fig. 1.**

---

## Author Comment (AC1) · 12 Oct 2016

The review by Dmitry Divine is in general positive, but contains two major comments.

The first major comment refers to the fact that when we deal with the smoothed series, the number of degrees of freedom should be reduced, which leads to less significant correlation and regression coefficients. This is also clearly demonstrated in the short comment by Thomas Laepple, who showed that two rows consisted of 52 random values and smoothed with a 27-year filter may occasionally exhibit high correlation coefficients (see also my reply to his comment). I agree with this remark, and will change the text accordingly. In particular, I will discuss the isotopic composition of precipitation as a parameter that "covaries with atmospheric circulation in a manner similar to temperature" (following Eric Steig) rather than simply a proxy of air temperature.

The corresponding changes will be made in Introduction. On the other hand, the link between the isotopic composition of precipitation and air temperature has strong and clear physical basis, which gives additional prove to the observed covariation (though statistically insignificant) between these two parameters in the studied region. Rather poor correlation between them may be explained by, first, "stratigraphic noise" in the ice core data and, second, by some climatological factors that may disturb the isotope-temperature relationship (see major comment 2). Thus, in the revised manuscript I will keep discussion of the quantitative temperature changes in the past in Section 3.3 and Conclusion, but with revised uncertainties. In particular, new estimate of the overall warming during the past 350 year is $1\pm0.6$ °C instead of $1\pm0.2$ °C.

The second major comment concerns the precipitation type, seasonality and the origin of moisture in the coastal and inland regions of the studied area. These factors may affect the observed relationship between snow/ice isotopic composition and air temperature, and thus should be discussed. Indeed, the coastal and inland areas of Antarctica are different in terms of precipitation regime: coastal sites receive relatively more moisture from high latitudes of Southern Ocean, and most of precipitation is snow from clouds, while inland sites receive relatively more moisture from lower latitudes, and much (or even most) of precipitation is "diamond dust" from clear sky. This may lead to biases in the isotope-temperature relationship, when the observed changes in the snow isotopic composition may be caused not only by the changes in local air temperature, but also, e.g., by changing conditions in the moisture source, or by changes in precipitation seasonality. Although these effects are widely recognized, they are usually not taken into account when interpreting the ice core data, simply because we do not know much about past changes in precipitation origin, type or seasonality. In our case, my opinion is that although the mentioned factors may play a role in the observed discrepancy between ice core record and instrumental temperature record in the PEL region, the influence of these factors is much less then the influence of the "stratigraphic noise". This can be clearly demonstrated by considering ice core records obtained in a short distance one from another (Ekaykin et al., 2014): even in this case

we still have relatively low correlation between individual ice core records and relatively low correlation of the stacked ice core series with the instrumental temperature record. I will add the corresponding discussion to the Section 3.3 of the manuscript.

Other comments

"Pages 2-3, Section 2.1: Q. on ice core dating. Did the authors use, wherever possible, counting the seasonal peaks in d18O to establish and/or support their core chronologies?" The counting of the seasonal peaks was only possibly for the "105 km" ice core, where it was used as the basis of the dating. In other records the seasonal signal is not preserved. I will change the text in order to make it clearer.

"Page 3, Line 16: The age uncertainty associated with the Nye model alone can also be estimated directly from the Nye formula, please see Divine et al., 2011 (Polar Research, 30, 7379, DOI: 10.3402/polar.v30i0.7379, on page 3) for details." Thank you for this comment, I will use this approach for independent estimate of the age uncertainty. In our case the main source of the uncertainty is the error of the accumulation rate, which gives the age uncertainty <10%. This figure confirms our estimate.

"Page 3, Line 29: ". . .to cut off the variability with periodicities lower than 27 years. . .". Use "shorter" rather than "lower". Please provide some more detail on the filtering procedure you have actually used." I used a rectangular-shaped filter that cut-off all the frequencies > 0.037 (i.e., periods < 27 years). I will add the corresponding information to the text.

"Page 5 Line 26. High correlation coefficient reported for AWS LGB59, is it based on 5 annual values only or the authors used the monthly means for this particular case? If the latter is correct did the authors subtract the annual cycle from the data?" Yes, the correlation between LGB59 with Vostok and Mirny is 0.95 and 0.96, but is only based on 5-year record. Although it is statistically significant with a 0,05 confidence level, I realize that the conclusion made on 5-year series does not look very solid. But I included this in the manuscript, since this information is supplementary (not main)

evidence that the climatic variability is uniform within the whole studied sector. Indeed, we have already demonstrated that climatic record at Vostok correlates with those at Mirny and Davis, so we may expect a high correlation between a point located in the middle of the sector with the mentioned sites.

"Page 5 Line 28: Just a comment: principal component analysis commonly used in climate sciences, could be considered a reasonable alternative to a cluster analysis" We agree that PC analysis could be used as well, but in this case we prefer to use the cluster analysis as it gives the result in a simple and intuitively understandable way.

"Page 8 Lines 3-5: since the presented slope estimate is based on the low-pass filtered series, a decreased number of DOF needs be taken into account. The STD on the estimated slope is presently underestimated and should be corrected; some more details on the method the uncertainty of the slope was calculated should be provided too." In our case, it was not possible to derive the isotope-temperature slope directly from the regression of the PEL2016 stacked series with the instrumental temperature record, since PEL2016 consists of normalized values. Thus, to calculate the isotope-temperature slope we used well-known relationship: slope $(y,x) = r(y,x) * std(y)/std(x)$. where $std(x)$ is the STD of temperature record, and $std(y)$ is the mean STD of individual isotope records As an estimate of the uncertainty of the slope, we used the uncertainty of the mean STD value of individual isotopic records (as indicated in Page 8, Line 3). But this estimate does not take into account the uncertainty of the correlation coefficient. So, the revised value of the isotope temperature slope will be $9\pm6$ ‰$°$C.

"Page 9 Line 23: "…the IOD is expected to affect the inland Antarctic climate…" can the authors provide any relevant reference pointing to a link between IOD and cyclonic activity in the coastal Antarctica?" The heat and moisture is brought to Antarctica by cyclones, this is why we suggested that the correlation between isotopic content of precipitation and IOD could be due to modulation of cyclonic activity by IOD mode. But so far we could not find a proof of it in literature (which does not necessarily means that our supposition is wrong), this is why we used air pressure at the coastal stations

as a rough proxy of cyclonic activity.

"Page 10, Line 4: A similar divergence in the longer term trends in d18O and accumulation was also observed for the coastal DML (see Divine et al., 2009, JGR,114, D11112, doi:10.1029/2008JD010475 ) but not on the plateau where both d18O and SMB showed positive trends (Altnau et al., 2015)." I will include this into discussion.

"Figure 5: please use different colors for 5b. The lines are difficult to discriminate with the presently used color palette. Correct the uncertainty interval on the reconstruction by adjusting for the number of DOFs." I will change the figure accordingly.

I agree with the other comments and will correct the text accordingly.

---

## Author Comment (AC2) · 12 Oct 2016

The short comment of Thomas Laepple supports the major comment by Dmitry Divine pointing out the underestimated uncertainty of the correlation between the smoothed series. By a simple test Thomas demonstrates how a false correlation may appear in the smoothed series.

I conducted similar test that confirmed the Thomas's conclusions. According to my calculations, the probability to get correlation coefficient = 0.66 between two random 52-year rows smoothed with a 27-year filter is 27 %. However, we should also take into account that the PEL2016 series is a stack of 2-5 individual series (see Figure 5a). If we take this into account in the tests, then the probability to get occasionally r = 0.66 is reduced to 17 %, but still it means that the observed correlation between

PEL2016 and the instrumental temperature record is insignificant. Thus I will make the corresponding changes in the manuscript (see also the answer to Reviewer 1).

I will also clearly indicate a poor correlation between our climate reconstruction and the previous reconstructions. This finding underline the fact that our ability to reconstruct the past climate on the centennial time-scale based on ice core data is very limited.

---

## Author Comment (AC3) · 12 Oct 2016

"Page 4, Line 21 – are all the correlations done on de-trended data?" No, but in these series the variance related to trend is significantly less than variance related to the short-term variability. We also tested the correlation on the de-trended series: interestingly, in this case the correlation is stronger. It means that on the short-term scale the temperature records are closely related than on the decadal scale (as discussed in section 3.1 and shown in Figure S2).

"Page 5, Line 11 – not sure if this was a mistake but should PDO be IPO? You are justifying the use of IPO because of a previous teleconnection with IPO?" We wanted to say that previously we found the relationship between the Vostok climate record and PDO, this is why we decided to check the link between the PEL2016 and PDO. But

instead of PDO we took IPO, as it should better work for the Southern Hemisphere. I will re-write this part of text to make the idea clearer.

"Somewhere in this section is would be good to include reference to the moisture source regions or airmass transport routes. Has any backtracjectory work been done in this region that you could reference? This might aid the discussion about the differences between stations?" This comment agrees with a similar comment by Referee 1. I plan to add a short discussion of moisture origin, precipitation type and seasonality to Section 3.3. As for the moisture source regions, the corresponding information could be taken from Sodemann H. and Stohl A. (2009) Asymmetries in the moisture origin of Antarctic precipitation. Geophys. Res. Lett., 36(22), L22803 (doi: 10.1029/2009GL040242).

"Page 8, Line 20 – can you add a short description of the little ice age? E.g. Cold period observed in northern hemisphere? I am a little nervous about defining LIA periods for Antarctic records. The pages 2k paper you cite states "There were no globally synchronous multi-decadal warm or cold intervals that define a worldwide Medieval Warm Period or Little Ice Age". Concluding that "a cold period is observed at approximately the same time interval as the little ice age reported in other regions" may be safer." I will change the text as you suggest.

"Page 8, Line 21 – Just for interest and comparison we also see a cold phase during the 1840s in the isotope record from Ferrigno (coastal Ellsworth Land). Might add evidence to it being a continental scale event. Thomas, E. R., T. J. Bracegirdle, J. Turner, and E. W. Wolff (2013), A 308 year record of climate variability in West Antarctica, Geophys. Res. Lett., 40, doi:10.1002/2013GL057782" Thank you, I will add this into discussion.

"Page 9 - Snow accumulation variability. This section is lacking information on the thinning functions applied to the records. You mention the Nye model was used for the 400km core but nothing about the 105 and 200km records. Please just specify which thinning method was used in the text." For the 105th and 200th km we also calculated

[Figure]

thinning using the Nye model. We also introduced the advection correction needed to take into account spatial changes of accumulation rate and snow isotopic composition upstream from the drilling site. I will add the corresponding information to the text.

"Table 1 – Suggest "this study" instead of "this work" For the sample resolution can you give an indicator of the number of samples per year? Or per decade for 400 km?" I will add this information.

"Figure 1 – Just a style issue but I found it hard to see the ice core locations on my screen. Consider changing the orange used." I suggest a new version of the figure (see supplement to this comment).

I agree with the other comments and will make the corresponding changes in the text.

[Figure]

Fig. 1.

---

## Author Response (AR1)

Reply to Dmitry Divine

**Major comments**

1) My first major comment concerns the method the authors used to estimate the isotope to
temperature gradient and its STD on the smoothed data. More specifically, it is not demonstrated that a
reduced number of degrees of freedom (DOF) in the data due to smoothing is taken into account. The
same applies to significance of the correlation coefficients reported for the smoothed series. For a 27-
year low pass filtered instrumental series of a length of about 60 years one have to expect about 5
independent data points only, implying that a simple sample variance (or STD) of the slope presented in
the manuscript is a biased estimator of an underestimated true variance. For a very simplified case of
AR(1) model of serial correlation in the data, taking the effect of autocorrelation into account to
estimate the confidence intervals (CI) on the slope estimate was summarized in Nychka et al., 2000
(available from
http://citeseerx.ist.psu.edu/viewdoc/download;jsessionid=A30C325B3A1E36EAB30B126EF74F974E?doi
=10.1.1.33.6828&rep=rep1&type=pdf ). To reassess the significance of correlation coefficients simple
adjustment for a number of independent samples in the t-distribution quantile can be applied as a
simplistic remedy of the problem.

We agree with this comment. The text was changed as following:

Section 3.3.:

The stacked dD record (built from low-pass filtered individual records)  is now compared with the
filtered PEL temperature composite (Fig. 4a). We observe a positive correlation with r = 0.66. **Although**
**the length of the series is 52 years, the number of degree of freedoms is only 4, due to the 27-year**
**filtering. The uncertainty of the correlation is ± 0,4, so it is statistically insignificant (p = 0,17)**.

We also modified the error bars in Figure 5b according to the larger uncertainty of the isotope-
temperature slope.

2) Some discussion on precipitation types/seasonality, and moisture origin that can be different for
the coastal and inland locations in the study area would be highly relevant in the context of the
observed discrepancies between the core series and the instrumental data.
We added the following text in Section 3.3.:

**This invokes a discussion of the factors that may disturb the correlation between the local air**
**temperature and the stable water isotopic composition of precipitation in Antarctica (Jouzel et al.,**
**2003).**

**Firstly, isotopic composition of precipitation is not a function of local air temperature, but of the**
**temperature difference between the evaporation area and the condensation site, which defines the**
**degree of heavy water molecules distillation from an air mass. The study of the moisture origin for**
**this sector of Antarctica (Sodemann and Stohl, 2009) demonstrates that different parts of the PEL**
**differ in their moisture origin. Coastal areas receive moisture from higher latitudes (46-52° S) and**
**from more western longitudes (0-40° E) than inland areas (34-42° S and 40-90° E). It means that even if**
**our sector is climatically uniform, as was shown above, the temporal variability of the precipitation**
**isotopic content may differ in the different parts of the sector due to varying moisture origin.**

**Secondly, we should define which temperature is actually recorded in the isotopic composition of**
**precipitation. For central Antarctica, where much (or most) of precipitation is "diamond dust" from**
**clear sky (Ekaykin, 2003), the effective condensation temperature is conventionally considered equal**
**to the temperature on the top of the inversion layer. But it is definitely not true for the coastal areas,**
**where most precipitation falls from clouds. Thus, the difference between near-surface and**
**condensation temperature may vary in space and time.**

**Thirdly, the precipitation seasonality is another factor that may change the relationship between the**
**air temperature and stable isotope content in precipitation. At Vostok the precipitation amount is**
**evenly distributed throughout the year (Ekaykin et al., 2003), so the snow isotopic content**
**corresponds well to the mean annual air temperature, but we don't have robust information neither**
**about the other parts of the PEL, nor about the seasonality changes in the past.**

**Yet we believe that the main factor that affects the isotope-temperature relationship is the**
**"stratigraphic noise". Indeed, even when we study the ice cores obtained in a short distance one from**
**another (Ekaykin et al., 2014), the correlation between the individual isotopic records is still small,**
**despite the same climatic conditions.**

**This is why we argue that constructing the stacked isotopic record is an optimal way to reduce the**
**amount of noise in the series and to highlight the variability that is common for the whole studied**
**region, provided that the region is climatically uniform.**

**Other comments**

Page 1 last line: "the only source of climatic data". Please use "primary" instead; there are
alternative though sparse sources of instrumental data such as earlier expeditions to Antarctica,
observations from ships logbooks etc.

Done

Page 2 Line 5: "…moreover unevenly distributed…", ".reflecting heterogeneous efforts…", "still
remain white spots". Awkward sentences, please check the language.

We changed the text:

The network of ice core records spanning the last centuries is **distributed highly unevenly**. A quite extensive coverage of some regions of Antarctica, such as West Antarctica (Kaspari et al., 2004) or Dronning Maud Land (Altnau et al., 2015;Oerter et al., 2000) contrasts with other regions that **still remain poorly studied**.

Page 2, Line 15: "Classically" can be omitted

Done

Page 2, Line 29: "…down to a 150 m depth…"

Done

Page 2: "Individual records" can be modified to "ice core data"

Done

"Pages 2-3, Section 2.1: Q. on ice core dating. Did the authors use, wherever possible, counting the seasonal peaks in d18O to establish and/or support their core chronologies?"

The counting of the seasonal peaks was only possibly for the "105 km" ice core, where it was used as the basis of the dating. In other records the seasonal signal is not preserved. We added the following text in order to make it clearer:

**This core is the only one where accumulation rate allowed the annual layers to be preserved in the snow thickness, so** the core was dated by layer counting.

"Page 3, Line 16: The age uncertainty associated with the Nye model alone can also be estimated
directly from the Nye formula, please see Divine et al., 2011 (*Polar Research*, 30, 7379, DOI:
10.3402/polar.v30i0.7379, on page 3) for details."

We added the following text:

**The uncertainty of the dating, estimated with the Nye model, mainly comes from the error of the**
**accumulation rate estimate and is evaluated as about 10 %.**

Page 3, Line 27: "…values were reduced in terms of mean and STD…". Awkward sentence, better to
refer to the procedure as a "mean and variance adjustment" or a "variance scaling" (see e.g. Esper
etal., 2005, GRL 32, L07711, doi: 10.1029/2004GL021236.

Page 3, Line 27: "…to avoid an artificial dominating…", please check the language

We changed the text as follows:

As for the short series (NVFL-3 and PV-10), they were normalized over 1978-2009 period, and then **the**
**mean and variance of** the normalized values were **adjusted to those of the long series** for the
corresponding period **of time**, in order to avoid **an overestimated contribution** of the short records in
the stacked series.

"Page 3, Line 29: "…to cut off the variability with periodicities lower than 27 years…". Use "shorter"
rather than "lower". Please provide some more detail on the filtering procedure you have actually
used."

We changed this text accordingly:

We then applied **a rectangular-shaped** low-pass filter to cut off the variability with periodicities **shorter**
than 27 years (**i.e., frequencies > 0.037**).

Page 4, Line 2: "…due to a very low SNR…" …and non-temperature effects on isotopes in
precipitation including post-depositional alterations.

Done

Page 4, Line 4: "…despite (some) common features…

Done

Page 4, Line 8: "…observed discrepancies do not arise from chronological uncertainties alone…"

Done

Page 4, Line 9: "…significant level of noise event in the filtered series". …and other than the ambient
temperature -related controls on the isotopic composition of precipitation.

Done

Page 4 Section 2.3. Subsection title can be changed to "Instrumental temperature data"

Done

Page 4 Line 17. "The data are available from…". Please mention explicitly that the annual means
were constructed from the monthly means.

Done

Page 5 Line 2: "…considered as a prevailing mode of atmospheric circulation in the SH representing
about 35% of the extratropical SH climate variability".

Done

Page 5 Line 2: "The monthly AAO index is available from…"

Done

Page 5 Line 22: "…to assess whether uniform climate variability pattern is monitored…". Awkward
sentence, consider revision.

We changed the text as follows:

Here, we first consider the surface air temperature recorded at the meteorological stations in
Princess Elisabeth Land, to assess whether **the studied sector is characterized by** uniform climate
variability, and to provide a reference regional temperature record for comparison with the ⬚D
stacked record.

"Page 5 Line 26. High correlation coefficient reported for AWS LGB59, is it based on 5 annual values
only or the authors used the monthly means for this particular case? If the latter is correct did the
authors subtract the annual cycle from the data?"

Yes, the correlation between LGB59 with Vostok and Mirny is 0.95 and 0.96, but is only based on 5-
year record. Although it is statistically significant with a 0,05 confidence level, I realize that the
conclusion made on 5-year series does not look very solid. But I included this in the manuscript,
since this information is supplementary (not main) evidence that the climatic variability is uniform
within the whole studied sector. Indeed, we have already demonstrated that climatic record at
Vostok correlates with those at Mirny and Davis, so we may expect a high correlation between a
point located in the middle of the sector with the mentioned sites.

Page 5 Line 27: "…that the region encompasses between these 3 stations…". Please check the
language and consider revision.

Corrected

"Page 5 Line 28: Just a comment: principal component analysis commonly used in climate sciences,
could be considered a reasonable alternative to a cluster analysis"

We agree that PC analysis could be used as well, but in this case we prefer to use the cluster
analysis as it gives the result in a simple and intuitively understandable way.

Page 6 Line 14: "…have a 30-year periodicity…". Due to a shortness of the data being analyzed,
referring to a "quasi-periodic variability" would be more appropriate. Mind also the edge effects of
any filtering procedure that in the zone of influence equal to a filter length at a specified timescale.

We changed the text as follows:

Both Vostok and Mirny **demonstrate a qasi-periodical variability with a period of about 30 years**

Page 7 Line 5: "…reflects a larger pressure gradient…"

Done

Page 7 Line 15: please see my major comment 1.

I re-estimated the significance of the correlations and changed the text accordingly:

However, different results emerge when considering the low-pass filtered time series. At multi-
decadal time scales, a strong positive correlation (r = 0.8, **significant with a 0,06 confidence level**)
relates PEL temperature and the AAO (Fig. 4a and 4b), and a very strong positive correlation
appears between PEL temperature and the IOD index (r = 0.93, **p < 0,05**).

"Page 8 Lines 3-5: since the presented slope estimate is based on the low-pass filtered series, a
decreased number of DOF needs be taken into account. The STD on the estimated slope is presently
underestimated and should be corrected; some more details on the method the uncertainty of the
slope was calculated should be provided too."

In our case, it was not possible to derive the isotope-temperature slope directly from the regression of
the PEL2016 stacked series with the instrumental temperature record, since PEL2016 consists of
normalized values.

Thus, to calculate the isotope-temperature slope we used well-known relationship:

slope (y,x) = r(y,x) * std(y)/std(x).

where std(x) is the STD of temperature record, and std(y) is the mean STD of individual isotope records

As an estimate of the uncertainty of the slope, we used the uncertainty of the mean STD value of
individual isotopic records (as indicated in Page 8, Line 3). But this estimate does not take into account
the uncertainty of the correlation coefficient. So, the revised value of the isotope temperature slope will
be 9±6 ‰/°C. We changed the text accordingly, and also modified the error bars in Figure 5b.

"Page 9 Line 23: "…the IOD is expected to affect the inland Antarctic climate…" can the authors
provide any relevant reference pointing to a link between IOD and cyclonic activity in the coastal
Antarctica?"

The heat and moisture is brought to Antarctica by cyclones, this is why we suggested that the
correlation between isotopic content of precipitation and IOD could be due to modulation of
cyclonic activity by IOD mode. But so far we could not find a proof of it in literature (which does not
necessarily means that our supposition is wrong), this is why we used air pressure at the coastal
stations as a rough proxy of cyclonic activity.

"Page 10, Line 4: A similar divergence in the longer term trends in d18O and accumulation was also
observed for the coastal DML (see Divine et al., 2009, JGR,114, D11112, doi:10.1029/2008JD010475
) but not on the plateau where both d18O and SMB showed positive trends (Altnau et al., 2015)."

We added the following text into the manuscript:

**Similar divergence of the centennial trends of snow isotopic composition and accumulation rate**
**was observed by Divine et al. (2009) at the coastal sites of Dronning Maud Land, but not at the**
**inland sites (Altnau et al., 2015).**

Page 10, Line 27: "…suggested to modulate…"

Done

Page 11, Line 8: please provide STD on the estimated slope.

Done

Page 11, Line28: "field technicians" or "field engineers" would be a more appropriate term

Done

Page 12, Line 1. "…in the framework…" , please indicate what abbreviation "LIA" stands for

Done

"Figure 5: please use different colors for 5b. The lines are difficult to discriminate with the presently
used color palette. Correct the uncertainty interval on the reconstruction by adjusting for the
number of DOFs."

The figure was modified accordingly.

Reply to Elisabeth Thomas

**General comments:**

Page 2 Line 1- some might consider borehole or historical records. Perhaps reword to

"primary" or "a valuable source"

Done

Line 13 – word missing "we find evidence of : : :", or "we observe a : : :"

This part of text is re-written completely:

We **note a not perfect correlation between the stacked isotopic record and regional surface air**
**temperature variations, underlying the fact that the isotopic content of precipitation is not simply a**
**proxy of temperature, but rather a parameter that covary with the local climate in a manner similar to**
**temperature (Steig et al., 2013)**.

Page 4, Line 4 – suggest remove "clear"

Done

Page 4, Line 8 – suggest replace "only" with "solely"

Done

"Page 4, Line 21 – are all the correlations done on de-trended data?"

No, but in these series the variance related to trend is significantly less than variance related to the short-
term variability. We also tested the correlation on the de-trended series: interestingly, in this case the
correlation is stronger. It means that on the short-term scale the temperature records are closely related
than on the decadal scale (as discussed in section 3.1 and shown in Figure S2).

Page 5, Line 1 - I know you are choosing to use the term AAO but perhaps an "also known as the SAM" would be helpful. The structuring of this paragraph could be improved.

Consider using "the AAO index is available from NOAA (include web link in brackets) and the British Antarctic Survey"

We modified the text as follows:

AAO index, **also known as SAM (Southern Annular Mode)**, is defined as a mean latitudinal difference
of sea level pressure at 40 ºS and 65 ºS, and is considered as a prevailing mode of Atmospheric
circulation in the Southern Hemisphere **representing about 35% of the extratropical SH climate**
**variability** (Marshall, 2003). The **monthly** AAO index is available **from NOAA**:

http://www.cpc.ncep.noaa.gov/products/precip/CWlink/daily_ao_index/aao/monthly.aao.index.b79.curre
nt.ascii.table (since 1979) **and** British Antarctic Survey (http://www.antarctica.ac.uk/met/gjma/sam.html)
since 1957, although data for the 1957-1978 period is considered to be less robust.

"Page 5, Line 11 – not sure if this was a mistake but should PDO be IPO? You are justifying the use of IPO because of a previous teleconnection with IPO?"

We wanted to say that previously we found the relationship between the Vostok climate record and PDO,
this is why we decided to check the link between the PEL2016 and PDO. But instead of PDO we took
IPO, as it should better work for the Southern Hemisphere.

I re-wrote this part of text:

We use IPO data because in the previous study **we found** a teleconnection between **the climate**
**variability in the central Antarctic and tropical Pacific** (Ekaykin et al., 2014).

Page 5, Line 15 – reference to SOI that is not defined in the text

We added the full name of SOI.

Page 5, Line 22 – suggest changing "monitored" to "observed"

We changed the text as follows:

Here, we first consider the surface air temperature recorded at the meteorological stations in Princess
Elisabeth Land, to assess whether **the studied sector is characterized by** uniform climate variability, and
to provide a reference regional temperature record for comparison with the dD stacked record.

Page 5, results and discussion

"Somewhere in this section is would be good to include reference to the moisture source regions or airmass transport routes. Has any backtracjectory work been done in this region that you could reference? This might aid the discussion about the differences between stations?"

We added the following text to the Section 3.3.:

**This invokes a discussion of the factors that may disturb the correlation between the local air**
**temperature and the stable water isotopic composition of precipitation in Antarctica (Jouzel et al.,**
**2003).**

**Firstly, isotopic composition of precipitation is not a function of local air temperature, but of the**
**temperature difference between the evaporation area and the condensation site, which defines the**
**degree of heavy water molecules distillation from an air mass. The study of the moisture origin for**
**this sector of Antarctica (Sodemann and Stohl, 2009) demonstrates that different parts of the PEL**
**differ in their moisture origin. Coastal areas receive moisture from higher latitudes (46-52° S) and**
**from more western longitudes (0-40° E) than inland areas (34-42° S and 40-90° E). It means that**
**even if our sector is climatically uniform, as was shown above, the temporal variability of the**
**precipitation isotopic content may differ in the different parts of the sector due to varying moisture**
**origin.**

**Secondly, we should define which temperature is actually recorded in the isotopic composition of**
**precipitation. For central Antarctica, where much (or most) of precipitation is "diamond dust"**
**from clear sky (Ekaykin, 2003), the effective condensation temperature is conventionally considered**
**equal to the temperature on the top of the inversion layer. But it is definitely not true for the coastal**
**areas, where most precipitation falls from clouds. Thus, the difference between near-surface and**
**condensation temperature may vary in space and time.**

**Thirdly, the precipitation seasonality is another factor that may change the relationship between**
**the air temperature and stable isotope content in precipitation. At Vostok the precipitation amount**
**is evenly distributed throughout the year (Ekaykin et al., 2003), so the snow isotopic content**
**corresponds well to the mean annual air temperature, but we don't have robust information neither**
**about the other parts of the PEL, nor about the seasonality changes in the past.**

**389** **Yet we believe that the main factor that affects the isotope-temperature relationship is the**
**390** **"stratigraphic noise". Indeed, even when we study the ice cores obtained in a short distance one**
**391** **from another (Ekaykin et al., 2014), the correlation between the individual isotopic records is still**
**392** **small, despite the same climatic conditions.**

**393** **This is why we argue that constructing the stacked isotopic record is an optimal way to reduce the**
**394** **amount of noise in the series and to highlight the variability that is common for the whole studied**
**395** **region, provided that the region is climatically uniform.**

**396**

**397** "Page 8, Line 20 – can you add a short description of the little ice age? E.g. Cold period

**398** observed in northern hemisphere? I am a little nervous about defining LIA periods for

**399** Antarctic records. The pages 2k paper you cite states "There were no globally synchronous

**400** multi-decadal warm or cold intervals that define a worldwide Medieval Warm

**401** Period or Little Ice Age". Concluding that "a cold period is observed at approximately

**402** the same time interval as the little ice age reported in other regions" may be safer."

**403**

**404** We changed the text as follows:

**405**

**406** A colder period is identified in 1750-1860 - **i.e., approximately at the same time interval as the** "Little
**407** Ice Age" **reported in the other regions** (PAGES 2k network, 2013).

**408**

**409** "Page 8, Line 21 – Just for interest and comparison we also see a cold phase during the

**410** 1840s in the isotope record from Ferrigno (coastal Ellsworth Land). Might add evidence

**411** to it being a continental scale event. Thomas, E. R., T. J. Bracegirdle, J. Turner, and E. W. Wolff (2013),
**412** A 308 year record of climate variability in West Antarctica, Geophys. Res. Lett., 40,
**413** doi:10.1002/2013GL057782"

**414**

**415** We changed the text accordingly:

**416**

**417** This minimum was also identified in an Antarctic temperature stack record (Schneider et al., 2006) – see
**418** Fig. 5d, as well as an ice core drilled in the Ross Sea sector (Rhodes et al., 2012) **and in the isotope**
**419** **record from Ferrigno (coastal Ellsworth Land) (Thomas et al., 2013)**.

**420**

**421** "Page 9 - Snow accumulation variability. This section is lacking information on the thinning

**422** functions applied to the records. You mention the Nye model was used for the

400km core but nothing about the 105 and 200km records. Please just specify which thinning method was used in the text."

We added the following text at the end of Section 2.1:

We also use the accumulation data from the site "200 km" (Fig. 1), spanning the period 1640-1987, as
published in (Ekaykin et al., 2000). **The accumulation values from sites "150 km" and "400 km"**
**were corrected both for layer thinning with depth and for the advection of ice from upstream of the**
**glacier to account for the spatial gradient of the snow accumulation rate**.

Page 10, Line 18 – suggest changing "has evidenced" for "demonstrates"

Done

Page 11, Line 10 - suggest changing "evidenced" for "observed"

Done

"Table 1 – Suggest "this study" instead of "this work" For the sample resolution can you give an indicator of the number of samples per year? Or per decade for 400 km?"

The Table was modified accordingly

"Figure 1 – Just a style issue but I found it hard to see the ice core locations on my screen. Consider changing the orange used."

The figure was corrected accordingly

[revised manuscript text omitted]

---

## Author Response (AR2)

Dear Dr. Stenni,

We have corrected the citation Jones et al., in press to Jones et al., 2016 throughout the whole text and in the References. We have also modified the citation PAGES_2k_network, 2013 to PAGES 2k Consortium, 2013 throughout the whole text and in the References.

Commas for separating decimals have been replaced by dots as notified.

The manuscript has been revised by a native English speaker, appropriate corrections have been implemented.

Kind regards,

Alexey Ekaykin on behalf of author's group